# A Multiplexed, Tiled PCR Method for Rapid Whole-Genome Sequencing of Infectious Spleen and Kidney Necrosis Virus (ISKNV) in Tilapia

**DOI:** 10.3390/v15040965

**Published:** 2023-04-14

**Authors:** Shayma Alathari, Dominique L. Chaput, Luis M. Bolaños, Andrew Joseph, Victoria L. N. Jackson, David Verner-Jeffreys, Richard Paley, Charles R. Tyler, Ben Temperton

**Affiliations:** 1Department of Biosciences, University of Exeter, Stocker Road, Exeter EX4 4QD, UK; d.chaput@exeter.ac.uk (D.L.C.); l.bolanos-avellaneda@exeter.ac.uk (L.M.B.); vlnj201@exeter.ac.uk (V.L.N.J.); c.r.tyler@exeter.ac.uk (C.R.T.); 2Centre for Environment, Fisheries and Aquaculture Science (Cefas), The Nothe, Barrack Road, Weymouth DT4 8UB, UK; andrew.joseph@cefas.gov.uk (A.J.); david.verner-jeffreys@cefas.gov.uk (D.V.-J.); richard.paley@cefas.gov.uk (R.P.); 3Sustainable Aquaculture Futures Centre, University of Exeter, Exeter EX4 4QD, UK

**Keywords:** *Oreochromis niloticus*, ISKNV, Artic-Network, aquaculture, long-read sequencing

## Abstract

Tilapia farming is one of the most important sectors in aquaculture worldwide and of major importance to global food security. Infectious spleen and kidney necrosis virus (ISKNV) has been identified as an agent of high morbidity and mortality, threatening tilapia aquaculture. ISKNV was detected in Lake Volta, Ghana, in September 2018 and spread rapidly, with mortality rates between 60 and 90% and losses of more than 10 tonnes of fish per day. Understanding the spread and evolution of viral pathogens is important for control strategies. Here, we developed a tiled-PCR sequencing approach for the whole-genome sequencing of ISKNV, using long read sequencing to enable field-based, real-time genomic surveillance. This work represents the first use of tiled-PCR for whole genome recovery of viruses in aquaculture, with the longest genome target (>110 kb dsDNA) to date. Our protocol was applied to field samples collected from the ISKNV outbreaks from four intensive tilapia cage culture systems across Lake Volta, between October 2018 and May 2022. Despite the low mutation rate of dsDNA viruses, 20 single nucleotide polymorphisms accumulated during the sampling period. Droplet digital PCR identified a minimum requirement of template in a sample to recover 50% of an ISKNV genome at 275 femtograms (2410 viral templates per 5 µL sequencing reaction). Overall, tiled-PCR sequencing of ISKNV provides an informative tool to assist in disease control in aquaculture.

## 1. Introduction

Nile tilapia (*Oreochromis niloticus*) is a key fish species for freshwater aquaculture, with a global production estimated at 4,525,400 tonnes [1], providing food, employment, as well as domestic and export earnings to large populations worldwide [2,3]. Tilapia production has almost doubled over the past decade [1], due to their relative ease of farming, marketability and stable prices [4]. Tilapia aquaculture provides an important source of nutrition, especially for populations that are dependent on a narrow range of staple foods. In Ghana, nearly 70,000 metric tonnes of tilapia were produced in 2018 [5], rising rapidly from only 954 tonnes in 2005 [6]. Most production in Ghana is conducted under high density stocking in floating cage systems, and is centred around Lake Volta (Figure 1), with hatcheries predominantly located besides the River Volta, below the dam to the lake [6]. Intensification of production in aquaculture is associated with risks of disease emergence and spread, as high stocking density and the number of reported viral outbreaks has increased steadily over the last few decades, resulting in catastrophic losses to fish farmers globally [7,8]. Although, the major disease agents are predominantly bacterial infections, such as Streptococcal infections [9], there is an increasing global burden of emerging viral infections, such as tilapia lake virus (TiLV), which has been a causative agent of high cumulative mortalities estimated at (80–90%) in farmed tilapia in Israel, Ecuador and Colombia [10].

Infectious spleen and kidney necrosis virus (ISKNV) is a Megalocytivirus, and one of five genera within the Iridoviridae family of large, enveloped, double stranded DNA viruses [12]. ISKNV virions are icosahedral, around 150 nm in diameter and contain a single linear dsDNA molecule 111,362 bp in length, whose structure is highly methylated at cytosines in the CpG and circularly permuted during infection [13,14]. The host range of ISKNV was previously considered to be narrow: extended surveys did not detect ISKNV in 18 fish species, including tilapia [15]. However, ISKNV has been observed in mandarin fish (*Siniperca chuatsi*) [14] and large-mouth bass (*Micropterus salmoides*) [16]. In 2012, an ISKNV outbreak occurred in tilapia in the United States with a 50–75% mortality rate over a two-month period [17]. In late 2018, unusual patterns of very high mortality in the Asutsuare region of Ghana (Figure 1, Farm 1) were reported in intensive tilapia cage culture systems across Lake Volta, with ISKNV confirmed as the likely causative agent through PCR and DNA sequencing of the major capsid protein. Samples collected from the same farms had all tested negative for the virus the previous year. A week following the first report, a second farm located in the Akuse region reported similar mortalities. By the end of 2018, despite the attempts to reduce losses by increasing the production of fingerlings, or treatment with antibiotics, most tilapia farmers in Lake Volta were not able to contain these mortalities. By mid-October, the Dodi region and the Asikuma region reported 10 tonnes of fish losses per day [11].

To mitigate the effects and spread of viral diseases in aquaculture, it is critical to achieve rapid detection of the causative agent, understand their epidemiology and disseminate the information efficiently to raise awareness [18]. Analysis of outbreaks in viral aquatic diseases in particular requires methods that offer a high level of strain discrimination [19]. Understanding the phylogeography of a viral outbreak provides vital information for containment, source identification and prevention of future outbreaks, yet current practices for epidemiological tracing focus on partial fragments of the MCP gene, do not inform changes occurring in other structural proteins or identify mutation sites on other relevant proteins that may alter vaccine development regions and/or changes in virulence. Whole genome sequencing (WGS) has revolutionised our ability to track infectious disease outbreaks by providing greater resolution of emerging diversity, allowing rapid and accurate identification of virulence factors of pathogens [20,21]. However, historically the long lead times and requirement for expensive sequencing infrastructure have limited application of WGS to understanding disease phylogeography.

The Artic-Network pipeline is an end-to-end system for generating real-time WGS epidemiological information, coupling a tiled-PCR approach to portable sequencing devices from Oxford Nanopore, enabling rapid deployment in resource-limited settings. This approach was successfully deployed to determine the phylogeography of Ebola and Zika outbreaks, as well as global surveillance of emerging variants of SARS-CoV-2 [22,23,24,25,26]. To date, the Artic-Network pipeline has been used for short, rapidly evolving RNA viruses (19 kbp for Ebola and 30 kbp for SARS-CoV-2). Using a similar approach to understand the phylogeography of large dsDNA viral genomes, which require many more tiled PCR products and evolve more slowly, limiting the emergence of novel variants, had not been tested. Here, we developed a protocol for WGS sequencing for real-time surveillance of ISKNV, optimising primers to recover ~96% of the ISKNV genome using the Artic-Network tiled PCR approach. When applied to samples from infected tilapia from Lake Volta, we were able to detect the accumulation of mutations across the ISKNV genome, during the sampling period. Successful field testing in Ghana showed that our method could be deployed for real-time surveillance as a field diagnostics tool. We confirm that the Artic-Network protocol can be adapted for long dsDNA viruses to provide useful phylogeographic information for managing disease outbreaks in aquaculture and beyond. A minimal viral load to recover >50% of the ISKNV genome (at least 50% of the nucleotides from ISKNV genome represented at >20-fold coverage in the sequence data) using tiled-PCR was established using droplet digital PCR (ddPCR) [27], to guide future sequencing efforts.

## 2. Materials and Methods

### 2.1. Samples and DNA Extraction

A total of 36 tissue samples from the spleen, liver and brain were collected from Nile tilapia (*Oreochromis niloticus)* from seven different fish farms in Ghana (Figure 1), during outbreaks of ISKNV. A total of 31 samples were collected by Cefas, from 10 October 2018 to 11 July 2019. An additional five samples were collected from a more recent outbreak in May 2022 (Table 1). Samples were stored in RNAlater^®^ and shipped to Cefas Weymouth Laboratory for processing. Samples collected from farms labelled 1 and 2 were washed twice in 750 µL of sterile 1× PBS to remove the RNAlater^®^ and homogenised using Matrix A and the FastPrep-24™ apparatus. Total nucleic acid was extracted using nanomagnetic beads (Genesig Easy DNA/RNA Extraction Kit, Primerdesign, Southampton, UK). For farms labelled 3 to 7, RNAlater^®^ was removed and the tissue samples weighed. Depending on the weight of the tissue available, tissue samples were diluted in RLT buffer (Qiagen, Manchester, UK) at either 1:10 *w/v* or a 1:5 *w/v*, pooled and homogenised per fish using Matrix A and the FastPrep-24™ apparatus to homogenise the tissues (MP Biomedicals, Eschwege, Germany). Following homogenisation, samples were diluted further with RLT buffer for a 1:60 *w/v* homogenate and clarified by centrifugation for 10 min at 3000× *g*. Total nucleic acid was extracted from 300 μL of the clarified sample using the EZ1&2 RNA Tissue Mini Kit without DNase (Qiagen, Manchester, UK) and eluted in 60 μL of RNAse-free water. DNA extraction for samples collected in May 2022 was performed using the DNeasy Blood and Tissue kit (Qiagen, Manchester, UK). A starting sample of 10 mg of pooled organs (kidney and spleen) were collected and dried for 5 min prior to DNA extraction using the manufacturer’s protocol. Eluted nucleic acid was stored at 4 °C for one week until processing.

### 2.2. Design of Primers

Primers to produce 2 kbp amplicons with an overlap of 50 nt were generated with PrimalScheme (v 1.3.2) [23], using ISKNV reference sequence (accession Number: AF371960.1). A total of 62 primer pairs spanned the full ISKNV genome, and the version (v1) was designated to this set of primers (Appendix A). As an initial development and testing of our methodology for genome recovery we used viral samples collected from the 2019 outbreak in Ghana (Farm 7) and propagated in Bluegill fry BF-2 cell lines (American Type Culture Collection, ATCC CCL 91) at Cefas [11]. Template DNA was recovered (total nucleic acid kit (ThermoFisher, Heysham, UK), extracted in a Maxwell^®^ RSC Instrument (Promega, Southampton, UK)). Each primer pair was individually tested against the template DNA, following the nCoV-2019 sequencing protocol v3: https://www.protocols.io/view/ncov-2019-sequencing-protocol-v3-locost-bh42j8ye?step=6 (accessed on 29 August 2020). PCR was performed with Q5 Hotstart High-Fidelity Polymerase (NEB) as follows: 98 °C for an initial heat activation for 30 s, 15 s at 98 °C for denaturation, followed by a 65 °C for annealing and extension step for 5 min for 30 cycles, and amplicons were visualised by gel electrophoresis. Four out of the 62 primer pairs failed to produce a product of appropriate size and were replaced by newer primers (v2) generated from a sequence alignment produced in Geneious Prime^®^ 2021.1.1 from the following sequences: Accession numbers (NC_003494, MT128666, MW 273354, MW273353). Finally, the (v3) primer set contained the v1 primers with additional alternate primers for drop-out regions. All primers can be found in Appendix A.

### 2.3. Primer Pool Preparation

Two primer pools were generated (Pool A and Pool B), containing odd and even numbered genomic regions, respectively, at 15 nM concentration per primer. Template DNA concentration was increased from 2.5 ng to 7.5 ng (freshly diluted viral DNA in nuclease-free water (NFW)). Two pools were prepared with alternating primer sets, as previously described [23].

### 2.4. Failed Regions Recovered Using Neighbouring Pairs

To determine whether variation from the reference sequence was responsible for failure of four primer pairs, we generated larger amplicon products to span these regions, using neighbouring primers to generate a 6 kb product which spanned the drop-out regions (Appendix A).

### 2.5. Library Preparation, Sequencing Protocols and Bioinformatic Processing

Using ISKNV cell line extracts, 2 kb amplicons were generated and pooled for sequencing using a FLO-MIN 106 (R9.4.1) (Oxford Nanopore Technologies, Oxfrord, UK) MinION flow cell. Library preparation was conducted using the Ligation Sequencing kit 1D (SQK-LSK109) (ONT) and Native Barcoding system (EXP-NBD104) (ONT), according to the manufacturer’s instructions, and following the Native barcoding amplicon protocol (version NBA_9093_v109_revD_12Nov2019). Amplicons were quantified using the Qubit dsDNA broad range kit (Invitrogen, Waltham, MA, USA), and pools A & B for each sample were combined and assigned a single barcode per sample. The equimolar amounts of each barcoded sample were pooled and taken forward for the adaptor ligation step using a total volume of 60 µL of DNA. An amount of 5 µL of Adaptor Mix II (AMII), 25 µL of Ligation Buffer (LNB) and 10 µL of T4 DNA Ligase were all added to the barcoded DNA. The reaction was incubated for 10 min at room temperature, and a 0.5× AMPure XP bead clean-up was performed, followed by 2 × 250 μL of SFB (ONT) washes. The pellet was resuspended in 15 µL of Elution Buffer (EB) for 10 min at 37 °C. An amount of 15 µL of the elute was retained and ~1 µg of adaptor ligated DNA was taken forward for priming and loading onto the flow cell.

Sequencing was run for 23 h. High accuracy base calling was carried out using the Oxford Nanopore Guppy tool (v. 4.0.15). Adapter trimming was performed, and samples were demultiplexed using guppy_barcoder. Reads below 1800 bp in length and above 2200 bp were removed. Reads were mapped to the reference genome from the NCBI (NC_003494) [14] using minimap2 (v.2.17, parameters: -x map-ont) [28]. Genome coverage is visualised in Tablet [29].

### 2.6. Construction of the Full ISKNV Genome Infecting Tilapia in Ghana

A complete reference ISKNV genome from the Ghana outbreak was reconstructed in a three-step protocol. First, consensus genomes were constructed separately using 2 kb amplicons and 6 kb amplicons. The 6 kb amplicons were individually amplified as they failed to amplify with a multiplex PCR. The consensus sequences of 6 kb and 2 kb were aligned using LASTZ v1.02, with default parameters in Geneious Prime^®^ 2021.2.2, revealing a gap spanning the region between primers 46 and 48. A separate amplicon library generated only using these primers was sequenced as above, and the sequences were aligned to the 2 kb/6 kb genome to close the gap. To recover the ends of the ISKNV genome, an amplicon library was generated from the last primer (62 f) and the first primer (1 r), sequenced as above and manually aligned to the constructed genome. All constructed consensus genomes used had a minimum of 20× coverage of the genome. The ISKNV consensus was annotated using Prokka version 1.13 (Seemann, T. Prokka: Rapid Prokaryotic Genome Annotation. Bioinformatics 2014, 30, 2068–2069.) [30]. Prokka was run with following parameters: --addgenes --compliant --kingdom Viruses. Predicted single nucleotide polymorphisms (SNPs) were assigned to the corresponding genes in Geneious.

### 2.7. Droplet Digital PCR to Determine Minimal Input for Genome Recovery of ISKNV Using the Tiled PCR Protocol

We first established the number of ISKNV templates required in a sample to recover at least 50% of the ISKNV genome at 20-fold coverage for robust error correction, using tiled PCR. Triplicates of 10-fold serial dilution from 6 ng to 6 × 10^−6^ ng of ISKNV from cell line extracts were used as a standard curve. Quantification of template strands in each dilution was performed in accordance with the manufacturer’s instructions (Bio-Rad, Watford, UK). The reactions included 10 μL of 2× ddPCR™ Evagreen (Bio-Rad, Hercules, CA, USA), 1 μL of each forward primer (5′ CGCCTTTAACGTGGGATATATTG 3′) and reverse primer (5′ CGAGGCCACATCCAACATC 3′) (200 nM) [31], and 8 μL of DNase/RNase-free H_2_O and 1 μL of DNA template. PCR amplification was performed with an initial step of 95 °C for 5 min, followed by 40 cycles of 95 °C for 30 s, 54.6 °C for 60 s and 1 cycle of 4 °C for 5 min, 1 cycle of 90 °C for 10 min, followed by 12 °C of 10 min. Microdroplets from each well were read using a QX200 Droplet Reader (Bio-Rad, WAtford, UK). The copy number of each well was evaluated by QuantaSoft™ version 1.2 (Bio-Rad, Watford, UK).

Serial dilutions from the above were sequenced on a single MinION flow cell following library preparation using a Ligation Sequencing kit 1D (SQK-LSK109) (ONT) and Native Barcoding system (EXP-NBD196) (ONT), according to the manufacturer’s instructions, and following the Native barcoding amplicon protocol: version NBA_9093_v109_revD_12Nov2019. The percentage of genome covered was estimated by at least 20-fold coverage (for consensus sequence polishing). A linear regression model (*genome recovery @ >20× coverage ~→log_10_ (number of template strands per µL*)) was used to determine the number of viral particles to achieve at least 50% recovery of the genome. ddPCR was also employed to detect the number of ISKNV templates present in samples collected from Farms 3–5, as these samples failed to amplify by tiled PCR. A positive control of 20 ng/µL ISKNV and similar conditions were followed, as mentioned above.

### 2.8. Epidemiology and Phylogeographic Analysis of ISKNV

To investigate the origin and diversity of ISKNV in Ghana, we performed whole-genome alignment of 40 genomes of samples collected from Lake Volta and different ISKNV strains previously sequenced (Appendix A). Consensus genomes were aligned using Geneious Prime^®^ 2021.1.1. Specifically, sequences were aligned using MAFFT [32], and a phylogeny was reconstructed using IQ-Tree [33,34]. The consensus sequences generated from each sample collected from Lake Volta were aligned to previously sequenced ISKNV genomes available from the GenBank NCBI. GenBank accession numbers and host species are listed in Appendix A.

### 2.9. Tiled PCR for MinION Sequencing of ISKNV Directly from Samples Collected from Lake Volta Outbreak

Samples from the Lake Volta ISKNV outbreak were processed using the Ligation Sequencing kit 1D (SQK-LSK109) (ONT) and Native Barcoding system (EXP-NBD104) (ONT), according to the manufacturer’s instructions, and following the Native barcoding amplicon protocol, as described in detail above. Nucleic acid extracts of ISKNV, along with a negative control, were pooled for sequencing, and MinION FLO-MIN 106 (R9.4.1) and flongle (FLO-FLG001) runs were performed. Sequencing was performed for 48 h for the MinION and ~24 h for the flongle. Super high accuracy (SUP) base calling was carried out after sequencing using the Oxford Nanopore Guppy tool (v. 6.0.1). Read demultiplexing was enabled by requiring barcodes for both ends, and reads below 1800 bp in length and above 2200 bp were removed. The Artic network pipeline was used to generate the consensus sequences for each genome. The workflow can be found in Appendix A. Augur bioinformatics toolkit (version 3.0.6) [35] (github.com/Nextstrain/augur) was used to process the genomes. Consensus genomes were aligned using MAFFT [32], and a phylogeny was reconstructed using IQ-Tree [33]. The tree was further processed using augur translate and augur clade to assign clades to nodes and to integrate phylogenetic analysis with metadata. Augur output was exported and visualised in auspice (github.com/Nextstrain/auspice) [34].

## 3. Results

### 3.1. Tiled PCR Recovers near Complete ISKNV Genome from Cell-Line Extracts

The full ISKNV genome was generated with MinION sequencing of ISKNV harvested from cell lines. Tiled-PCR products using the v1 primer scheme generated 192,317 reads, with a median read length of 1942 bp, and yielded ~75% genome recovery of the ISKNV genome, when aligned to the reference genome. All but four primer pairs were successful in generating 2 kb amplicons. Following this, primers generated from a sequence alignment (v2) of ISKNV ancestral genomes (listed in Appendix A), successfully amplified dropped regions when tested individually, and were used to replace the four failing primers, creating a newer primer version (v3). This version was used for all subsequent Lake Volta ISKNV samples.

To reconstruct the full ISKNV genome, 6 kb amplicons spanning the full genome were recovered, and the percentage of the genome with at least 20× coverage was 83.76%. These amplicons were combined with 2 kb amplicons to recover the full genome, end regions and primer pair 47 (a 6 kbp region that did not amplify within the pool, but did amplify separately). This reconstruction generated a near complete ISKNV genome spanning 99.79% of the ISKNV reference (NC_003494) with 99.82% average nucleotide identity and 19 ambiguous bases. A total of 137 SNPs were identified when compared with the ISKNV reference genome (NC_003494)—58 of these mutations were non-synonymous. Mutations were located in the putative ankyrin repeat protein (NP_612299.1), NTPase (NP_612285.1), DNA-directed RNA polymerase II (NP_612256.1) and thymidine kinase (NP_612254.1).

### 3.2. ddPCR Determined Minimal Input for Genome Recovery of ISKNV Using the Tiled PCR Protocol

To evaluate the optimal concentration of ISKNV needed for genome recovery using the tiled PCR method, we measured the number of ISKNV viral templates from 6 ng to 6 × 10^−6^ ng. A minimum of 10 template molecules of ISKNV (to 6 × 10^−5^ ng) were needed to recover any of the genome with the required per-nucleotide coverage of >20-fold for accurate error correction. Genome recovery increased logarithmically from 10 template strands to 10,000 template strands, where ~75% of the genome was recovered (Figure 2). The minimum requirement to recover 50% of an ISKNV genome was 275 fg (~2410 viral templates) in 5 µL of input DNA for each sequencing reaction. Figures generated by the QuantaSoft™ version 2.1 (Bio-Rad, Hercules, CA, USA) are found in Appendix A.

ddPCR was also applied for the field samples (farm 3–5) that failed to amplify using the tiled PCR protocol. Detecting the number of viral templates in samples collected from Farm 3 and 4 showed very low ISKNV concentration (<1 viral template), while samples collected from Farm 5 had a high concentration of ISKNV, with up to ~5000 viral templates per ng. According to Ramierez et al., July 2019, samples from Farms 4 and 5 were recovered from recent mortality events but had no remaining observable clinical disease [11].

### 3.3. Epidemiology and Phylogeographic Analysis of ISKNV Is Not Solely Related to Host Species

Whole genomes from previously published reference strains from different hosts were aligned with samples collected from Ghana, aligning with MAFFT v7.450 [32], in Geneious Prime (Figure 3). ISKNV within samples collected from Ghana belonged to a separate lineage compared to samples collected from other ISKNV outbreaks. The Brazilian strain ON212400.1, although also infecting Nile tilapia, seemed distantly related to samples collected from Ghana. ISKNV from tilapia samples in Ghana were most closely related to those from an outbreak in Albino sharks (MW273353), in the United States, and were least related to samples collected from mandarin fish (*Siniperca chuatsi*) and barramundi (*Lates calcarifer*). Host species are listed in Appendix A.

### 3.4. Phylogenetic Analysis Indicates Multiple Introductions of ISKNV in Fish Samples Collected from Lake Volta

Phylogenetic analysis of ISKNV within the Ghana outbreak of 2018–2022 was performed using Augur and visualised in Auspice (Figure 4). Initial outbreaks in Lake Volta clustered into four distinct clades, and each clade had a mix of samples from different farms. The three most closely related to the reference strain were identical and from three different farms, indicating possible multiple introduction events. The highest genome recovery was obtained from sample 6.2 at ~96%, with samples 2.2 and 2.11 having the lowest coverage, at 44% and 35% recovery, respectively (Appendix A), and the median genome recovery for all samples was 87.83% (85.61–88.63%, 95% CI, 1000 bootstraps). The consensus sequence of all the ISKNV samples obtained in this study displayed similar dropout regions in several locations of their genomes, with poorly recovered regions including: (1) a repeat region located at 23,273 bp to 23,768 bp; (2) between ORF014R and ORF018L; and (3) in the putative DNA polymerase (ORF025).

A total of 137 polymorphisms were observed when comparing samples from the first outbreak in Ghana, in 2018, to the ancestral strain (based on SNP-calling against reference genome NC_003494.1). Of those SNPs, 20 showed variations among samples taken in 2018 and 2022 (excluding the dropout regions). Four of the five samples taken in May 2022 from the Asikuma region (Farm 2) clustered with those taken at the same location in 2018, but they have diverged independently due to a non-synonymous SNP (T3934C) within the MCP. These were highlighted in Figure 4 as “Latest outbreak”. An additional mutation (C4328T) in the MCP was also unique to all Ghana samples compared to other outbreaks. A second SNP in the virulence gene ORF022L was also unique to the Ghana samples.

## 4. Discussion

ISKNV has caused major losses in aquaculture, with infections reported for more than 52 marine and freshwater species, and is continuously expanding to different continents [35]. To understand infectious disease dynamics in aquaculture, we have described the development and implementation of a new workflow to track viral outbreaks in fish using whole genome sequencing.

We report here the first tiled PCR that successfully generated near complete genomes of the large nucleocytoplasmic DNA virus ISKNV and its use to assess the epidemiology of an ongoing epidemic of ISKNV in Nile tilapia in Ghana. A full-length genome was initially obtained from 2 kb and 6 kb amplicons of ISKNV from cell culture isolates from early in the epidemic. Fifty-eight non-synonymous SNPs were identified relative to reference genome sequence (NC_003494). We observed a mutation in ISKNV thymidine kinase (TK), which has previously been correlated to increased neurovirulence and mortality of the host in another dsDNA virus—Herpes Simplex Virus (HSV-1) [36], and natural mutation in the TK gene of these viruses have been associated with an increase in drug resistance [37]. Thus, observed variations may in part explain the rapid spread of ISKNV early on in the outbreak, in conjunction with the naivety of the regional tilapia Akosombo strain to the disease. When compared to ancestral samples collected from different hosts, two mutations in the ankyrin repeat protein were only seen in the strains infecting tilapia fish in Ghana. This protein has previously been shown to play a role in modulating host range and cellular immune signalling [38]. A single mutation located in the ORF022L may have increased the virulence of ISKNV in Lake Volta. Zeng et al. have defined this part of the genome as a possible virulence gene and selected it as a target gene when constructing a gene deletion vaccine for ISKNV [39].

Sequencing viral material directly from 36 samples provided insight into the relatedness of viruses collected from four different fish farms, by examining their evolution in relation to geographical spread. ISKNV samples that only showed a positive result by nested PCR failed to amplify using our tiled PCR, despite some samples showing very high ISKNV concentrations when tested by ddPCR. This could be due to fragmented DNA of ISKNV samples collected from these farms with less than 2 kb fragments, or the presence of residual fragments of non-replicating ISKNV, as these farms witnessed a past mortality event and had no clinical signs of infection during the time of sampling.

We recovered high-quality (>72% complete) genomes of ISKNV from 31 out of 36 samples collected during the ISKNV outbreak in Ghana fish farms. Phylogenetic analyses showed patterns of similar haplotypes circulating both within and between farms, indicating a shared source of infection, possibly through epidemiological links such as movement of fish or equipment, including infected live fry and fingerlings for stocking purposes, water, wild and escaped cultured tilapia as vector reservoir, and potentially other vector species. The three most closely related samples to the reference strain were identical and from three different farms, confirming multiple introductions of ISKNV and/or rapid transmission across the farms. Samples taken from farm 2 were most closely related to farm 6, despite samples being collected after seven months. Moreover, there is evidence of the rapid mutation of ISKNV in Lake Volta following the first outbreak in 2018, in comparison to the previous evolutionary rate since the first documented ISKNV outbreak in 2001. The original probable index case for the introduction of this virus into the naïve population of tilapia in Lake Volta, Ghana, from Ramirez-Paredez et al. [11], was not accessible and not sampled, nor was the virus sequenced. Given the timing of subsequent disease events on various farms on the lake, most subsequent detections, isolations, and sequence data ([11] and this publication) are likely secondary re-introductions/movements.

ISKNV genomes from Ghana appear to include two polymorphisms in the major capsid protein, a standard target for single-gene phylogenies of this disease. Within the latest samples taken in May 2022, a new non-synonymous mutation in this region was observed and has not been identified previously in any ISKNV genomes to date, suggesting continued evolution of the outbreak and requiring further study.

Twenty polymorphisms were observed across the sampling period, within the recovered regions of the genomes. This number is likely to be an underestimate because of the several dropout regions across the sequenced genome and/or error correction of the DNA genome between sampling periods. Amplicon drop-offs are common, and usually affected by viral load, sample quality, and constant viral evolution, resulting in mutations on primer binding sites [40]. Therefore, despite examining the outbreak occurring across a short period of time, and ISKNV being a dsDNA virus, divergence was seen in samples collected from the farms under investigation, shedding some light on evolutionary origins in the phylogenetic analysis and confirming the utility of PCR-tiled approaches for viral phylogeography in large dsDNA viruses.

We did not attempt to evaluate intra-host variation of ISKNV in this study, and it is possible that the consensus sequences generated for each sample represent a flattening of true biological variability within samples. However, unlike RNA viruses, there are few reports of quasi-species within dsDNA viruses. The capacity to maintain a large viral genome without extinction through accumulated mutation is correlated to polymerase fidelity. Bacteriophages T2 and T4 are dsDNA viruses with similar genome size to ISKNV and have mutation rates of ~10^−8^ substitutions per nucleotide per replication cycle [41], approximately four orders of magnitude lower than the estimated mutation rate required to sustain a quasi-species population [42]. Therefore, it is likely that loss of intra-host variants within the consensus sequence of each sample would be minimal. In addition, the error-rate of individual Oxford Nanopore reads limits the capacity to discriminate between sequencing error and biological variability. Even with advanced methods for identifying intra-host SNVs (iSNVs), the false discovery rate of iSNVs using Oxford Nanopore data alone was ~55% in a rapidly evolving RNA virus, and is expected to be higher in viruses that evolve more slowly, such as ISKNV. In studies where quantification of iSNVs is required, replicated Illumina sequencing libraries per sample, combined with Nanopore data, is recommended for accurate quantification of iSNVs [43,44,45,46,47]. The increased costs and loss of field-based sequencing of this approach would need to be weighed against the likelihood and importance of detecting intra-host variability. Although not yet a day-to-day diagnostic tool for fish diseases, the method described here significantly reduces the cost of whole genome sequencing of important pathogens and makes it feasible in the field. Such information supports control strategies including the modelling of epidemiological links and potential vaccination or resistance breeding.

## 5. Conclusions

This work represents promising results with the potential to reveal a real-time view into the evolution and spread of ISKNV and other viral pathogens in aquaculture. This work here provides a platform from which it is feasible to replicate the Artic-Network “lab-in-a-suitcase” approach to disease tracking and management in aquaculture in remote and resource-limited locations. With appropriate training and guidance, this workflow represents a suitable framework for local authorities in lower- and middle-income countries to contain and track different viral diseases in their localities.

## Figures and Tables

**Figure 1 viruses-15-00965-f001:**
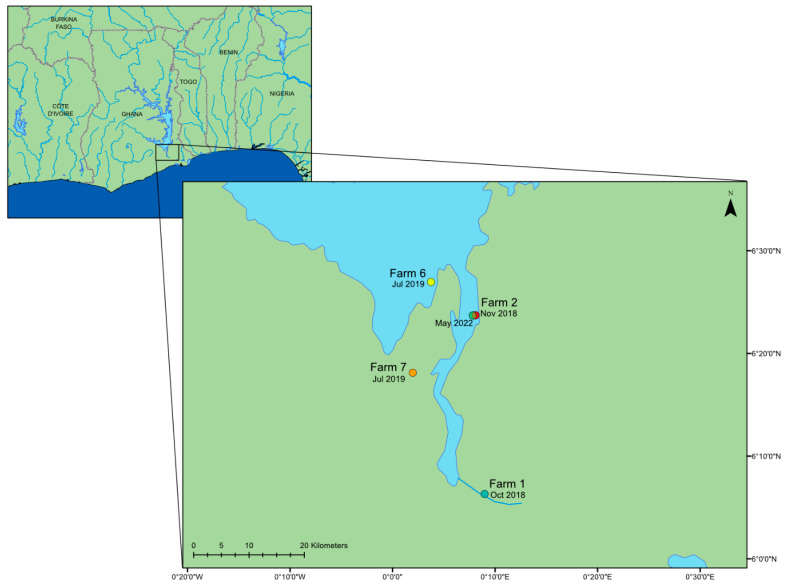
A map of the lower region of Lake Volta in Ghana, West Africa, showing the date and location of the farms where the outbreaks of mortality occurred; locations retrieved from [11]. This map was constructed using ArcGIS (GIS software). Version 10.0. Redlands, CA, USA: Environmental Systems Research Institute, Inc., 2010.

**Figure 2 viruses-15-00965-f002:**
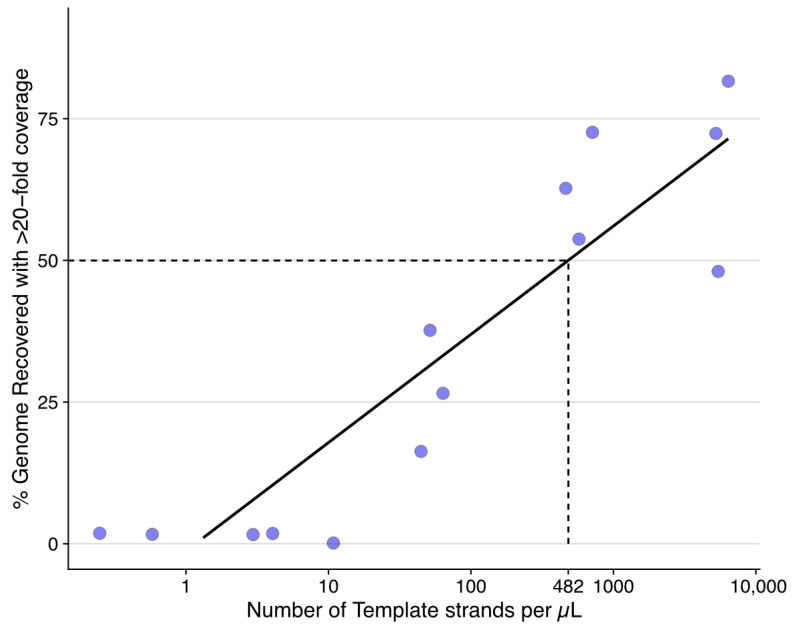
Successful recovery of >50% of the ISKNV genome required 482 template strands per µL (2410 viral templates per 5 µL sequencing reaction), with a minimum of 0.2 copies per µL to recover >0% of the genome with at least 20-fold coverage for error correction. Number of viral templates was measured using ddPCR from a serially diluted ISKNV template, which was subsequently sequenced and processed as described in the text.

**Figure 3 viruses-15-00965-f003:**
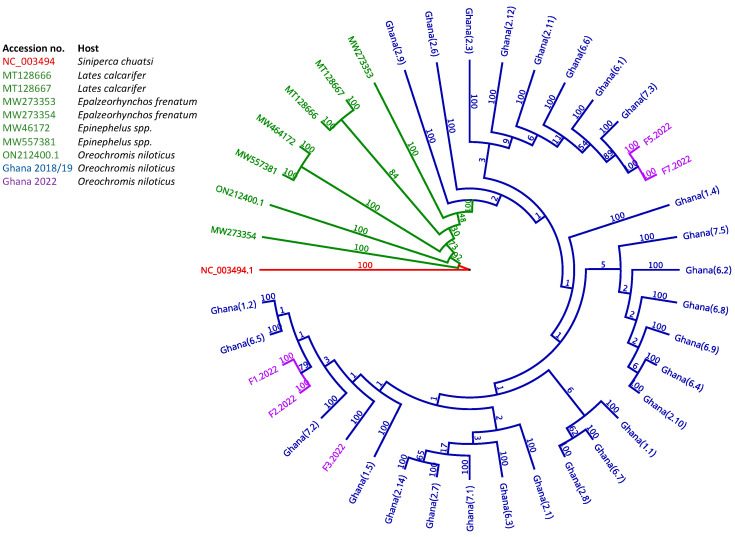
Phylogeny of whole ISKNV genomes of 36 samples collected from Lake Volta (2019 in blue and 2022 in purple) with whole ISKNV genomes reported in the GenBank, in green (listed in Appendix A), using MAFFT [32] with the bootstrapped branch support. The tree was rooted to the ISKNV reference genome (NC_003494), shown in red. Numbers in brackets after the Ghana samples from this study are in the format <*farm identifier*>.<*sample identifier*>.

**Figure 4 viruses-15-00965-f004:**
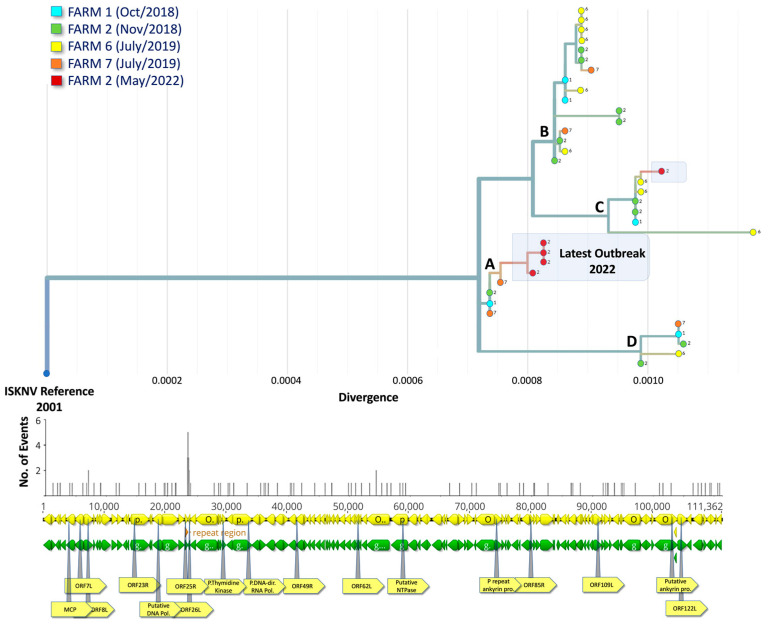
Phylogenetic placement of ISKNV genomes from Ghana and their associated farms. The horizontal axis indicates divergence relevant to the root of the tree. Clades are labelled A–D. The colour of the tips represents the date of sample collection; the number and location of mutational events are shown in the diversity panel below. Sequences from the latest outbreak in 2022 are highlighted.

**Table 1 viruses-15-00965-t001:** Dates and regions for collection of the 36 samples from four different farms in Lake Volta, Ghana. Data for Farms 3–5 have been described previously [11].

Farm	Number of Samples	Date	Region
**1**	5	18 October 2018	Dodi
**2**	11	28 November 2018	Asikuma
**2**	5	20 May 2022	Asikuma
**6**	10	10 July 2019	Dasasi
**7**	5	11 July 2019	Akosombo

## Data Availability

All data are deposited in NCBI BioProject ID: PRJNA935699. Accession number for the ISKNV strain infecting tilapia in Lake Volta in July 2019: OQ513807.

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
