# Peer review of "A Multiplexed, Tiled PCR Method for Rapid Whole-Genome Sequencing of Infectious Spleen and Kidney Necrosis Virus (ISKNV) in Tilapia"

_viruses, 2023, doi:10.3390/v15040965_

Round 1

Reviewer 2 Report

Present study developed a tiled PCR method for genome sequencing of ISKNV. The work is interesting, but many aspects of the MS need to be improved.

1.      Description about the tiled-PCR is not enough. It is better to provide a schematic diagram about the method used in the MS, which could make the MS clearer.

2.      The tiled-PCR and droplet digital PCR are two different PCR methods, which used different equipment. I don’t understand that the minimal input for genome recovery determined by droplet digital PCR can be applied in tiled-PCR.

3.      In the materials and methods section, all samples are collected from fish tissues, but “cell lines” are mentioned several times in the results section. Which cell lines were used in the study?

4.      “Genome recovery” was mentioned several times in the MS, but its definition was not provided.

5.      The font size in figure 3 and figure 4 is too small. What the purple font indicated is unknown in figure 3.

6.      The authors stated that the epidemiology of ISKNV is unrelated to host, but they also stated that “ISKNV from tilapia samples were least related to samples from mandarin fish and …”.

7.      There are negligence in reference citations, but I can’t point out the details because no line numbers in the MS.

Round 2

Reviewer 1 Report

- variations within the MinION data do not originate only from sequencing errors. They are also mutations due to the quasi-species nature of viruses. Therefore the intra-sample variability was not considered at all. It would have been interesting considering the focus on the SNPs (3934 or 4328).

- Please correct the latin name of mandarin fish 

- correct line 405 (and et al)

Author Response

We again thank the reviewer for their continued efforts to improve the manuscript. The reviewer raises the issue of quasi-species as a source for variation within the MinION reads in addition to sequencing error. We did not investigate intra-host variability and identification of iSNVs in the sequence data for two reasons. First, there are few reports of quasi-species in dsDNA viruses, and the estimated mutation rates of similarly sized dsDNA viruses are approximately four orders of magnitude lower than the estimates proposed by Eigen and Schuster to maintain a quasi-species. Second, while there have been attempts to do so in the literature, teasing apart real iSNVs from sequencing error in MinION data is extremely challenging, with a 55% false positive rate even with advanced methods. Thus, it was felt that iSNVs could not be confidently predicted in the samples. We have added text to the manuscript highlighting these issues and acknowledging that consensus sequences may be flattening real iSNVs within the samples.

The latin name for mandarin fish and line 405 have been corrected.